# Respiratory Distress Complicating Falciparum Malaria Imported to Berlin, Germany: Incidence, Burden, and Risk Factors

**DOI:** 10.3390/microorganisms11061579

**Published:** 2023-06-14

**Authors:** Bodo Hoffmeister

**Affiliations:** Department of Pulmonary Medicine and Infectious Diseases, Vivantes-Klinikum Neukölln, 12351 Berlin, Germany; bodohoff@gmx.de

**Keywords:** imported falciparum malaria, severe malaria, acute pulmonary oedema, acute respiratory distress syndrome

## Abstract

While European healthcare systems face resource shortages as a consequence of the coronavirus pandemic, numbers of imported falciparum malaria cases increased again with re-intensifying international travel. The aim of the study was to identify malaria-specific complications associated with a prolonged intensive care unit (ICU) length of stay (ICU-LOS) in the pre-COVID-19 era and to determine targets for their prevention. This retrospective observational investigation included all the cases treated from 2001 to 2015 at the Charité University Hospital, Berlin. The association of malaria-specific complications with the ICU-LOS was assessed using a multivariate Cox proportional hazard regression. The risk factors for the individual complications were determined using a multivariate Bayesian logistic regression. Among the 536 included cases, 68 (12.7%) required intensive care and 55 (10.3%) suffered from severe malaria (SM). The median ICU-LOS was 61 h (IQR 38–91 h). Respiratory distress, which occurred in 11 individuals (2.1% of the total cases, 16.2% of the ICU patients, and 20% of the SM cases), was the only complication independently associated with ICU-LOS (adjusted hazard ratio for ICU discharge by 61 h 0.24, 95% confidence interval, 95%CI, 0.08–0.75). Shock (adjusted odds ratio, aOR, 11.5; 95%CI, 1.5–113.3), co-infections (aOR 7.5, 95%CI 1.2–62.8), and each mL/kg/h fluid intake in the first 24 treatment hours (aOR 2.2, 95%CI 1.1–5.1) were the independent risk factors for its development. Respiratory distress is not rare in severe imported falciparum malaria, and it is associated with a substantial burden. Cautious fluid management, including in shocked individuals, and the control of co-infections may help prevent its development and thereby reduce the ICU-LOS.

## 1. Introduction

Due to its unique pathophysiology, various life-threatening complications may develop suddenly in falciparum malaria, even after the initiation of an effective therapy. Most fatalities occur during the first 48 h of admission [1,2]. Patients with one or more of such complications (the so-called severe malaria criteria, Table 1) should, therefore, be treated in an intensive care unit (ICU). In countries with low or middle income (LMICs) where the disease is endemic, however, these resources are often limited. In contrast, the European high-income countries (HICs) in which most imported cases occur, i.e., France, Great Britain, and Germany, have resource-rich healthcare systems. In the pre-COVID-19 era, Germany, for instance, provided more than 33 ICU beds per 100,000 population [3]. A higher standard of care in general and a better availability of ICU capacities in particular, were associated with a better outcome of falciparum malaria imported to European countries compared to LMIC in the past.

However, the coronavirus pandemic demonstrated that even the capacities in HIC can rapidly be overburdened. Its consequences continue to have a substantial impact on available resources. Shortages of medication, materials, and, most importantly, nurses, currently affect medical care in many institutions. While the number of malaria cases imported to European countries has sharply decreased as a result of the international travel ban during the pandemic [4], case numbers are expected to reach the pre-pandemic level soon with re-intensifying travel activity. The optimal treatment of imported falciparum cases is not only beneficial for those affected. It will also help reduce ICU-LOS, and thus relieve pressure on valuable ICU capacities. 

The present study aimed to identify the complications associated with prolonged ICU-LOS among patients with falciparum malaria imported to Berlin, Germany, in the pre-pandemic era and to determine targets for improved management.

**Table 1 microorganisms-11-01579-t001:** Definition of malaria-specific complications according to the WHO 2022 definition [5] with minor modifications and ARDS, according to the actual “Berlin definition” [6].

Criterion	Definition
Jaundice	Plasma or serum bilirubin > 50 µmol/L (>3 mg/dL) with parasitemia > 100,000/µL
Hyper-parasitemia	>10% parasitized erythrocytes
Decompensated shock	Systolic blood pressure < 80 mmHg with a need for norepinephrine dosages > 0.05 µg/kg/min to maintain a mean arterial blood pressure > 65 mmHg despite adequate hydration
Renal impairment	Plasma or serum creatinine > 265 µmol/L (>3 mg/dL) or blood urea > 20 mmol/L (>120 mg/dL)
Respiratory distress (APO and ARDS)	Oxygen saturation of room air < 92%, respiratory rate > 30/min, and bilateral opacities on chest imaging
Acute pulmonary oedema (APO)	Oxygen saturation of room air < 92% and respiratory rate > 30/min together with bilateral opacities on chest imaging
Acute respiratory distress syndrome (ARDS)	Lung injury within 1 week of admission with progression of respiratory symptoms; bilateral opacities on chest imaging not explained by other lung pathologies; respiratory failure not explained by heart failure or volume overload; PaO_2_/FiO_2_ ≤ 300 mmHg under a minimum PEEP of 5 cmH_2_O (applied using non-invasive or invasive ventilation)
Significant bleeding	Including recurrent or prolonged bleeding from the nose, gums, venepuncture sites, hematemesis, or melena
Severe malarial anemia	Hemoglobin level < 7 g/dL and/or hematocrit < 20% with parasitemia > 0.5%
Impaired consciousness	Glasgow coma scale (GCS) < 11
Acidosis	Base deficit > 8 mmol/L, and/or plasma bicarbonate < 15 mmol/L, and/or venous plasma lactate ≥ 5 mmol/L, or ≥45 mg/dL
Hypoglycemia	Blood glucose level < 2.2 mmol/L or <40 mg/dL
Multiple convulsions	>two convulsions within 24 h

Abbreviations: ARDS, acute respiratory distress syndrome; PaO_2_/FiO_2_, oxygenation index; PEEP, positive end-expiratory pressure; WHO, World Health Organization.

## 2. Materials and Methods

### 2.1. Data Collection

All the adult patients (≥18 years) with imported falciparum malaria that were hospitalized between 1 January 2001 and 31 December 2015 in the Charité, University Hospital, Berlin, were included (Figure 1). Some individuals had been treated more than once with falciparum malaria during the study period in the institution. Only their first episodes were included in the analysis. Among those patients requiring intensive care, the majority had been treated in the ICU of the Department of Infectious Diseases and Pulmonology under the guidance of a team of experienced infectious disease specialists. This team also supervised the therapy of the individuals cared for in other ICUs. All the ICUs employed the same electronic patient data management system (PDMS; COPRA5, COPRA System GmbH, Berlin, Germany), which allowed, inter alia, for continuous recording of vital signs and oxygen saturation as well as for electronic fluid balancing. The standardized electronic files containing data on the demographics; travel history; full medical history, including prior malaria episodes; current medication; and results of physical examinations and laboratory and radiologic investigations were available for all the patients. The study represents a secondary analysis of a previous investigation.

### 2.2. Clinical Management

The diagnosis of falciparum malaria relied on thin and thick blood smears. Parasitemia was expressed as the percentage of parasitized erythrocytes (1% corresponding to approximately 50,000 parasites/µL). Severe malaria was defined according to the World Health Organization (WHO) 2022 definition as the presence of ≥1 malaria-specific complication (Table 1) [5]. Minor modifications were made for the diagnosis of decompensated shock, which required the use of vasopressors, and for respiratory distress, which required radiologic proof of bilateral opacities in chest imaging, together with a respiratory rate of >30/min and an oxygen saturation of the room air < 92% (Table 1). Depending on whether a minimum PaO_2_/FiO_2_ ≤ 300 mmHg was recorded under the mechanical ventilation, applying a minimum positive end-expiratory pressure (PEEP) of 5 cmH_2_O respiratory distress was further discriminated into acute pulmonary oedema (APO) and the acute respiratory distress syndrome (ARDS; Table 1) [6]. Anti-malarial therapy was instituted as soon as possible after the establishment of the diagnosis. Prior to 2006, anti-malarial therapy of severe cases relied exclusively on quinine in combination with doxycycline or clindamycin. Quinine hydrochloride was given as an intravenous infusion (IV) over 4 h in an initial loading dose of 20 mg salt (i.e., 16.4 mg base) per kg ideal body weight (IBW), followed by 10 mg/kg IBW every 8 h starting 8 h after the loading dose (Table 2). After 48 h, 10 mg/kg IBW were administered every 12 h. Doxycycline was given 100 mg IV every 12 h and clindamycin was given 600–900 mg IV every 8 h. Artesunate only became available in Germany in 2007 (in the study center already in 2006). Since it was not manufactured in accordance with European good manufacturing practice and, hence, lacked market authorization, it was mainly used in patients with high parasitemia or with contraindications to quinine. Artesunate was administered with an initial dose of 2.4 mg/kg IBW IV followed by the same dose 12, 24, 48, and 72 h thereafter. If able to take oral medication, the patient then received a full course of oral artemether/lumefantrine or dihydroartemisinin/piperaquine. Supportive management consisted of a restrictive fluid management, renal replacement therapy (RRT, veno-venous hemofiltration), vasopressors (norepinephrine, started with a mean arterial pressure, MAP, <65 mmHg despite the fluid therapy that was considered sufficient for the actual clinical situation by the attending intensivists), and non-invasive and invasive mechanical ventilation where indicated. In order to assess the severity of the acute disease and the pre-existing chronic co-morbidity, the Simplified Acute Physiology Score II (SAPS II) on the admission as well as an age-adjusted Charleson co-morbidity index (CACCI) were calculated for all the patients [7,8].

### 2.3. Study Endpoints

The study’s primary endpoint was the ICU-LOS, which was measured in hours. The secondary endpoints were the risk factors for the development of complications associated with a prolonged ICU-LOS.

### 2.4. Statistical Analysis

The best way to model the ICU-LOS has been debated in the literature. With an ICU-LOS of <4 days, the Cox proportional hazard (CPH) regression was considered an appropriate approach [9]. The study site was a referral hospital for malaria patients. Since none of the patients had been transferred to other hospitals and none had died, there were no competing risks in the study population. Therefore, the conventional Cox proportional hazard regression with discharge after the median ICU-LOS (i.e., after 61 h) as the endpoint was considered to be the appropriate statistical approach. In a univariate analysis, the associations linking the individual malaria-specific complications to the instantaneous hazard ratio for discharge were assessed by building Cox proportional risk models with censoring of all the patients discharged after 61 h. To ensure the robustness of the results, sensitivity analyses with discharge after 38 and 91 h (i.e., after the first and the third quartile of the ICU-LOS) as the endpoints were also performed [10]. The best subset selection method was then used to identify the best combination of the covariates for a multivariate model [11]. Only the complications with significant associations in all three analyses were included in the final multivariate Cox proportional risk model with discharge after 61 h as the endpoint. The proportional hazard assumption was tested for the final model using the Schoenfeld residual test. The influential observations were tested by the dfbeta values. The statistical significance level for all the analyses was set at 5%.

For the identification of the risk factors, a logistic regression was used. The covariates were selected on the basis of a review of the recent literature [12,13,14,15,16,17]. Due to the limited sample size, however, sparse data bias was encountered when determining the odds ratios, even when penalized logistic regression methods such as the exact logistic regression or Firth’s method were employed [18,19]. In contrast to the frequentist methods, the Bayesian approach often performed better in data sets with small sample sizes [20]. Therefore, the risk factors were determined using a univariate Bayesian logistic regression employing R’s rstanarm package and using default (weakly informative) priors. Firth’s logistic regression was also performed as the sensitivity analysis for each covariate and for the final multivariate model, allowing for a comparison of the coefficients and medians of the posterior distribution. The best combination of the predictor variables for the final multivariate model was again selected by the best subset selection method. Prior information was included in the analysis where available. If no prior information was available, default (weakly informative) priors were used. The significance of the parameters was tested using the region of practical equivalence (ROPE) test, where values < 2.5% were considered as probably significant and values < 1.0% were considered significant [21]. The built-in model diagnostics included tests for the influential observations, normality of the residuals, and collinearity. The median of the posterior distribution with its 95% credibility interval (CI) was reported for the parameters with significant associations in the multivariate analysis. The effect existence was described by the probability of direction (pd; a parameter strongly correlated with the frequentist *p*-value; a pd-value of 97.5% corresponds approximately to a two-sided *p*-value of 0.05). 

For the statistical analysis, R version 4.1.0 (manufacture, city, if any state, country) was used (for the employed R packages, refer to Appendix A—Statistical Analysis). The findings were reported according to the strengthening the reporting of observational studies in epidemiology (STROBE) statement cohort studies checklist (Appendix A).

## 3. Results

### 3.1. Management of the Patients

A total of 558 imported falciparum malaria cases were treated during the study period in the institution, representing 7.1% of the total 7866 cases notified in Germany within that period. Twenty-two cases had to be excluded (Figure 1). Fifty-five of the remaining 536 cases (10.3%) retrospectively fulfilled the 2022 WHO definition of severe malaria. A total of 68 individuals (12.7%) required ICU admission. According to the 2022 WHO definition, 41 (60.3%) of these cases suffered from severe malaria (SM), while 27 (39.7%) patients had uncomplicated malaria (UM). Since the definition of severe malaria changed three times during the study period (i.e., in 2006, 2010, and 2015), not all the patients admitted to the ICU met the actual definition (Table 1). Fourteen cases with a severe disease were treated in general wards due to the sufficient clinical stability (11 of these cases originated from malaria-endemic countries). In some individuals, life-threatening conditions other than the criteria of severe malaria, such as advanced hyponatremia (Na^+^ < 115 mmol/L) or severe congestive heart failure, were responsible for ICU admission. The median number of malaria-specific complications in ICU admission, according to the WHO 2022 definition, was one (interquartile range, IQR 0; 2, range zero–nine). With the exception of renal impairment and metabolic acidosis, a clear correlation between these complications was not evident (Figure 2). Nearly half of the cases (*n* = 33, 48.5%) received artemisinin-based regimens. It was noted that all the patients were treated with a restrictive fluid management (Figure 3). The median fluid intake was 2.1 mL/kg/h (IQR 1.6–3.1, range 0.7–6.9 mL/kg/h) on day 1, 1.4 mL/kg/h (IQR 0.9–2.3, range 0.4–6.1 mL/kg/h) on day 2, and 1.3 mL/kg/h (IQR 1.1–1.7, range 0.3–5.0 mL/kg/h) over the rest of the ICU stay. Notably, the subgroup of ARDS patients received the highest fluid volumes throughout their ICU stay (Figure 3). Vasopressors were needed for 13 (19.1%) patients, and renal replacement therapy (RRT) was required in seven (10.3%) individuals for a median of 5 (IQR 4; 9) days. Co-infections were identified in a total of 19 (28.0%) cases (Table 3). Community-acquired co-infections were diagnosed in 14 (20.6%) patients, healthcare-associated co-infections occurred in eight (11.8%) individuals, and in two patients (2.9%) either type was found. One patient (1.5%) was diagnosed with two different community-acquired co-infections (*P. vivax* and hepatitis B). Respiratory distress complicated 11 cases and was diagnosed after a median of 17 h (IQR 2; 70 h, range 0–144 h). With a median of 17 h (IQR 2; 34 h), the six cases with APO were diagnosed earlier compared to the five ARDS cases that were diagnosed after a median of 51 h (IQR 15; 89 h). Low-flow oxygen therapy was sufficient to treat hypoxia in three (4.4%) patients with respiratory distress, while mechanical ventilation was required in eight (11.8%) individuals (non-invasive mechanical ventilation, NIV, *n* = 3, invasive mechanical ventilation, IV, *n* = 5). These eight individuals remained on mechanical ventilation for a median of 35 h (IQR 16; 150 h, range 2–384 h, total of 948 h on mechanical ventilation). Weaning these patients from mechanical ventilation was uncomplicated in all cases. All the patients survived to discharge.

### 3.2. Length of ICU Stay 

The length of the ICU stay ranged between 9 and 644 h with a median of 61 h (IQR 38–91 h) and a total of 6341 h (264 patient days). In the patients with SM, the ICU-LOS was significantly longer than in the UM cases (median of 66 versus 38 h, *p* < 0.0001). Three complications were found to be associated with a prolonged ICU-LOS in the univariate Cox proportional hazard regression: severe malarial anemia, decompensated shock, and respiratory distress (Table 4). The patients in whom respiratory distress had occurred had the longest ICU-LOS. This was especially true for the subgroup of the ARDS cases. Accordingly, respiratory distress was the complication with the strongest association to the ICU-LOS of all the malaria-specific complications in the univariate Cox hazard proportional regression. Respiratory distress, decompensated shock, and severe malarial anemia were also identified as the three best predictor variables for a multivariate model using the best subset selection method. These three variables were, thus, included in a multivariate Cox proportional hazard regression with discharge after 61 h as the endpoint. The model was adjusted for age, origin, and artemisinin treatment as the important potential confounders influencing the ICU-LOS. Respiratory distress was the only complication found to be independently associated with the ICU-LOS in this model (Table 5).

### 3.3. Risk Factors for Respiratory Distress

In order to identify the risk factors for the development of respiratory distress available in clinical practice, various socio-demographic, clinical, and laboratory parameters were selected on the basis of a literature review (Table 6). Fifteen of these covariates were associated with the development of respiratory distress in the univariate Bayesian logistic regression. The covariate history of or active malignancy was excluded from further analyses due to its small case number. The best combination of predictor variables for a multivariate model was then determined from the remaining 14 covariates using the best subset selection method. The three covariates, namely co-infections, decompensated shock, and fluid intake, on day 1 proved to be independently associated with the development of respiratory distress when included in a multivariate Bayesian logistic regression (Table 7). For shocked patients, an odds ratio for the development of ARDS of 2.16 was known from the literature. This information was, thus, included as prior information. For the two other covariates, no sufficient prior information was available. Hence, default (weakly informative) priors were used. The effect of shock had a 99.1% probability of being positive (median = 2.44, 95% credibility interval, 95%CI, (0.42, 4.73)) and could be considered as significant (0% in ROPE)). The effect of co-infections had a 98.35% probability of being positive (median = 2.01, 95%CI (0.15, 4.14)) and could be considered as significant (0% in ROPE)). The effect of each additional mL/kg/h fluid intake on day 1 of admission had a 98.72% probability of being positive (median = 0.77, 95%CI (0.11, 1.63)) and could also be considered as significant (0% in ROPE).

## 4. Discussion

The present study examined a cohort of imported falciparum malaria cases treated under favorable conditions. Modern, well-equipped ICUs provided nearly unlimited resources in terms of the diagnostics, medication, organ support, and trained medical stuff familiar with the unique pathophysiology of the disease. In this setting, respiratory distress proved to be the only malaria-specific complication independently associated with a prolonged ICU-LOS. The study emphasized the significance of controlling co-infections and avoiding fluid overload, even in shocked individuals, as these factors increased the risk of developing respiratory distress.

With an exponential growth in parasite biomass, the sequestration of infected (iRBC) and non-infected erythrocytes (RBC) in the microcirculation, and the intense systemic inflammation with generalized endothelial dysfunction, which could ultimately lead to endothelial barrier breakdown, falciparum malaria has a unique pathophysiology. The plasma concentrations of the biomarkers for endothelial activation, such as the von Willebrand factor (vWF) and angiopoietin-2 (ANG-2), are markedly elevated in severe malaria and have been associated with both the severity and mortality of the illness [22,23]. Marked upregulations of the vWF and ANG-2 have also been found in oedematous alveoli in post-mortem lung sections of Thai patients with malaria-associated ARDS (MA-ARDS) [24]. The pathogenesis of this life-threatening syndrome is still incompletely understood. Inflammation is thought to play the key role as it increases the alveolar–capillary permeability, leading to the extravasation of neutrophils, RBC, protein-rich plasma fluid, and ultimately, pulmonary oedema formation [25]. The complication occurs most commonly in adults lacking semi-immunity (e.g., travelers or residents of low transmission areas) and typically occurs alongside other complications, such as renal failure, cerebral malaria, or metabolic acidosis, carrying a poor prognosis [24,26]. In contrast to micro-vascular obstruction, which can be rapidly reversed by effective anti-malarials, generalized endothelial activation and systemic inflammation may persist for weeks (the so-called post-treatment inflammatory effect). As the lung is the organ system most vulnerable to the effects of increased capillary leakiness, respiratory distress can occur at any time during the course of the disease, even after a complete parasite clearance has been achieved [26,27,28]. The radiologic confirmation relies on a demonstration of bilateral opacities with or without pleural effusions in chest imaging. The main difference between APO and MA-ARDS lies in the severity of impaired oxygenation, which requires ventilatory support in the latter syndrome [26]. The epidemiologic data on MA-ARDS is scarce. The reported incidence rates in severe malaria vary widely between different studies and range from 2.1% to 29.1% [26,29]. Bacterial co-infections (e.g., pneumonia, sepsis), aspiration, and hypoalbuminemia are reported as contributing factors [25,26,29]. Fluid overload as a consequence of excessive intravenous fluid replacement, congestion secondary to heart failure, or fluid retention in renal failure, has also repeatedly been described as an aggravating factor [25,26,28,30]. Unfortunately, clinician recognition tends to be low, making ARDS an under-recognized and under-treated condition. In the Lung Safe Study, the diagnosis was correctly established in only 51% of mild and in 79% of severe ARDS cases [14].

With this background, the risk factors for the development of respiratory distress in imported falciparum malaria identified in the present study are plausible. Pulmonary infections and bacterial sepsis are generally the most common causes of ARDS. In the respective endemic areas, various *Plasmodium* species also play a causative role [31]. Numerous studies described the impact of co-infections, such as HIV or bacterial sepsis, on the course of malaria in general [32,33] and on the development of respiratory distress in particular [26,34,35]. However, upon closer inspection, few of these studies provide conclusive evidence that co-infections have an independent influence on the development of MA-ARDS. The present study examined a real-world scenario. The results demonstrated that managing falciparum malaria necessarily includes the management of a broad spectrum of community-acquired and hospital-acquired co-infections. On the basis of the multivariate analysis of the current study, an independent influence of these co-infections on the development of respiratory distress could be considered likely. Although severe imported malaria routine empirical treatment with broad-spectrum antibiotics cannot be recommended [36], the present study illustrated that a high degree of clinical vigilance, targeted microbiologic diagnostics, and timely initiation of empirical treatment are needed to achieve a favorable outcome.

Shock is another well-established risk factor for the development of ARDS. This association was also evident in the multivariate analysis of the present study. Both falciparum malaria and concomitant infections can cause or worsen shock. Fluid therapy is an essential component of shock management. While the recommended fluid volumes for initial resuscitation in patients with septic shock are approx. 30 mL/kg/h, the fluid volumes recommended for falciparum malaria are approximately ten times lower [37]. Due to its special pathophysiology, the dividing line between under- and over-hydration is known to be thin in falciparum malaria [5,28,30]. The validated tools for guiding the fluid therapy, however, are unavailable for clinicians. Therefore, the best possible clinical differentiation between sole malaria and putative co-infection is not only essential for a successful management but also for continuous and accurate fluid balancing. The usage of an ePDMS facilitates this. The review of the electronic charts revealed that the patient cohort examined in the present study had in fact been treated with a restrictive fluid management. However, the significantly higher fluid intake in the ARDS subgroup in the first 48 treatment hours suggested a causative role in the development of the complication. This assumption was supported by the results of the multivariate analysis of the current study as well as by previous investigations demonstrating the deleterious effects of a liberal fluid management in severe falciparum malaria [28,30]. In contrast, there is currently no evidence that a restrictive fluid management with an administration of 2–3 mL/kg/h worsens kidney function or tissue perfusion [37]. The fact that the median fluid intake in the subgroup of ARDS patients was <5 mL/kg/h on day 1 and <4 mL/kg/h on the second day after admission illustrated once again this thin dividing line.

These findings underscored that the risk factors leading to ARDS in other conditions were also associated with the development of respiratory distress in imported falciparum malaria. Importantly, the three risk factors identified in the present study were plausible individually as well as in their pathophysiologic interplay. Other factors, such as hypoalbuminemia or pre-existing chronic disorders, will likely also contribute to the development of respiratory distress in falciparum malaria. However, the present study was unable to define their roles, primarily due to the limited sample size.

In a recently published study of patients with severe imported falciparum malaria treated in a Portuguese ICU, the median ICU-LOS of the patients with ARDS was 13 days compared to 3 days for those without this complication [38]. These figures were close to the median ICU-LOS values for the patients with and without respiratory distress observed in the current study (11.5 versus 2.5 days). What are possible explanations for this striking association between respiratory distress and a prolonged ICU-LOS? While metabolic acidosis and coma are the leading causes of death in the first 48 h after admission in settings with limited medical resources, these complications can be survived with effective anti-malarials and high-standard intensive care (including invasive ventilation in comatose individuals). The long-lasting systemic inflammation in falciparum malaria, however, predisposes to typical consequences of intensive care in the aftermath, such as co-infections, hypoalbuminemia, and fluid overload, prolonging the ICU-LOS. Shock and mechanical ventilation are associated with critical illness myopathy, another well-known consequence of intensive care, which may aggravate the muscle weakness induced by the malaria itself [39]. Finally, respiratory distress usually does not occur separately, but most commonly together with other severe complications as part of multi-organ dysfunction and can, thus, be considered an indicator of a particular high-disease severity in general. This high degree of disease severity is reflected, for instance, by the significantly higher SAPS II scores in the ARDS cases compared to the individuals without the complication that have been observed in both the Portuguese study and the current study.

Similar to the other causes of ARDS, prevention may be the best medicine for respiratory distress in imported falciparum malaria [40,41]. However, staff shortages are currently having a significant impact on medical care in many European institutions. Thousands of nurses in Europe have quit their jobs since the beginning of the pandemic and many more are considering following their example. Shortages in nurse staffing have a negative impact on patient outcomes, such as mortality, readmissions, and the LOS [42]. It is, therefore, likely that these shortages will lead to prolonged ICU-LOSs in patients with imported falciparum malaria. The good news is, however, that improved medical education is a countermeasure with proven life-saving benefits [43].

The present study had several limitations, with the main limitation being its retrospective, single-center design and a long observation period that limited its generalizability. Few outcomes of interest were predisposed per se to sparse data bias. This inherent methodical problem was met by performing sensitivity analyses and by employing a Bayesian logistic regression, an approach known to yield robust results even in small data sets. The odds ratios determined for the individual risk factors may, nevertheless, still overestimate the true effects and should, therefore, be interpreted with caution.

However, the study also had several strengths. The setting of well-equipped ICUs provided close monitoring of the respiratory rates, pulse oximetry, and blood gases, as well as high-quality chest imaging. This allowed for exact and timely diagnoses as well as the accurate differentiation of APO and ARDS. The usage of the same ePDMS in all ICUs allowed for accurate fluid balancing. In the absence of validated tools for guiding fluid management, clinicians need concrete recommendations. The study provided these figures and, thereby, emphasizes the importance of meticulous fluid management in falciparum malaria. Finally, to the best of our knowledge, the study was the first to identify the risk factors for the development of respiratory distress in imported falciparum malaria cases.

## 5. Conclusions

The current study provided exact figures for the frequency of respiratory distress in a large cohort of imported falciparum malaria cases. The complication occurred in 2.1% of the total cases, 16.2% of the cases admitted to the ICU, and 20% among those with severe malaria. In a setting of nearly unlimited medical resources, respiratory distress proved to be the only malaria-specific complication independently associated with a prolonged ICU-LOS. The identified risk factors for the complication were the same associated with ARDS in other conditions. As such, they are likely, in large part, modifiable. A cautious fluid management, including in shocked individuals, and the best possible management of co-infections may help prevent the development of respiratory distress in imported falciparum malaria and thereby improve patient outcomes as well as relieve pressure on ICU capacities.

## Figures and Tables

**Figure 1 microorganisms-11-01579-f001:**
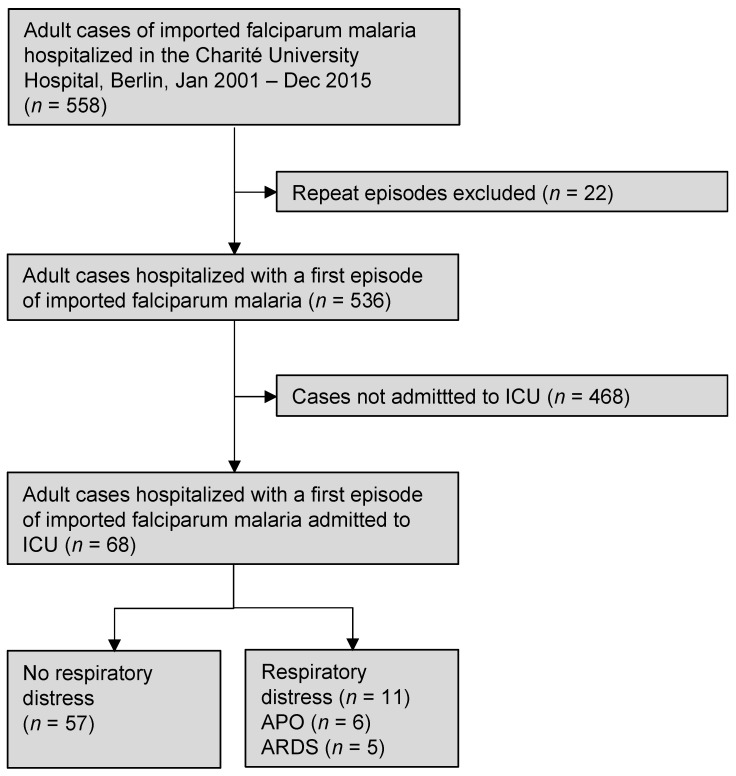
Flowchart of the patients included and excluded in the analysis. Abbreviations: APO, acute pulmonary oedema; ARDS, acute respiratory distress syndrome; ICU, intensive care unit.

**Figure 2 microorganisms-11-01579-f002:**
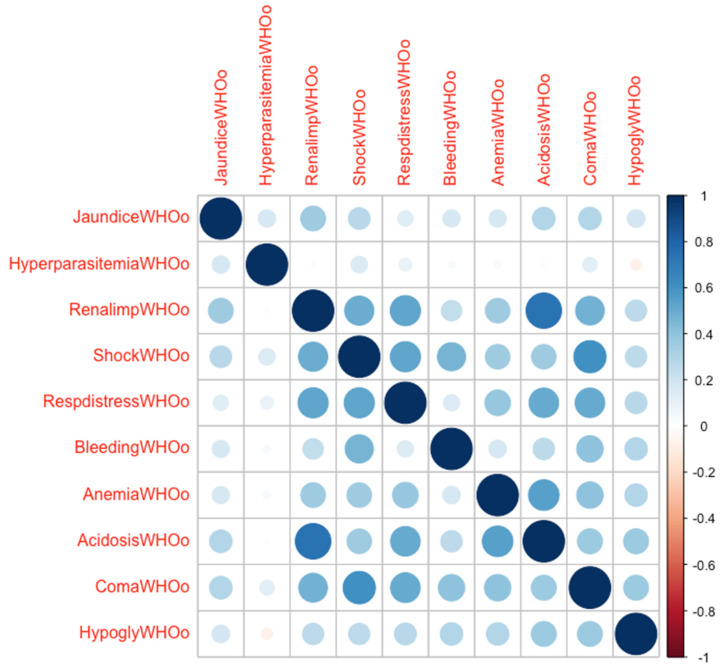
With the exception of renal impairment and metabolic acidosis (correlation coefficient, r, 0.73), a clear correlation between the various malaria-specific complications was not evident. In this correlation plot, the dot sizes and shadings symbolize the strength of the correlation, a positive association is colored in blue, a negative one in red. Respiratory distress occurred equally often together with shock and renal impairment (each r = 0.53) as well as with metabolic acidosis and coma (each r = 0.51).

**Figure 3 microorganisms-11-01579-f003:**
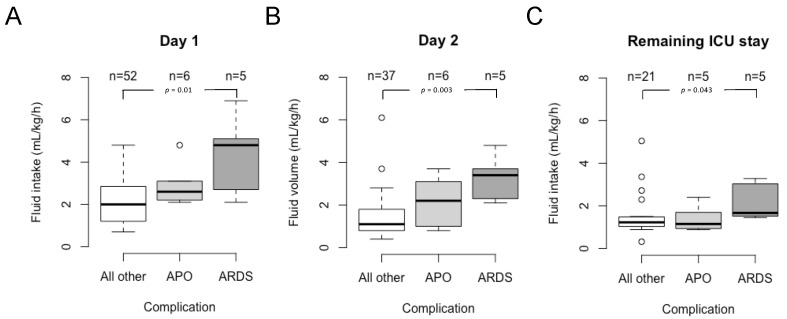
Overall, fluid management was restrictive in the patient cohort. However, the patients who developed MA-ARDS in the course of their disease had a significantly higher fluid intake over the whole of their ICU stay than the patients without respiratory distress. The panels depict the fluid intake on day 1 (**A**) and day 2 (**B**) after ICU admission as well as over the remaining ICU stay (**C**). Note that the case numbers per interval varied due to the continuous discharge of patients that had recovered from the ICU. In addition, body weight was not properly recorded in five patients, making the calculations of the fluid intake per kg bodyweight impossible. Abbreviations: APO, acute pulmonary oedema; ARDS, acute respiratory distress syndrome; ICU, intensive care unit.

**Table 2 microorganisms-11-01579-t002:** Anti-malarial treatment regiments used in 68 patients with imported falciparum malaria requiring intensive care.

Substance	Dosage	Duration	Median Parasitemia (IQR)	Number (%) of Cases Treated
Artemisinin derivates			8.0 (2.5; 10.3)	33 (48.5)
Artesunate	Loading dose of 2.4 mg/kg IBW intravenously	2.4 mg/kg IBW 12, 24, 48, and 72 h after loading dose. When able to take oral medication, the patients received a full course of oral artemether/lumefantrine or dihydroartemisinin/piperaquine	9.5 (5.1; 11.3)	25 (36.8)
Artemether/Lumefantrine	Four tablets of 20/120 mg orally	0, 8, 24, 36, 48, and 60 h after diagnosis	1.9 (0.7; 2.5)	6 (8.8)
Dihydroartemisinin/Piperaquine	Four tablets of 320/40 mg orally	0, 24, and 48 h after diagnosis	-	2 (2.9)
Quinine-based regimens (*n* = 35)			8.0 (4.3; 9.6)	35 (51.5)
Quinine/Doxycyline	20 mg quinine hydrochloride (16.4 mg base)/kg IBW loading dose intravenously. Doxycycline 100 mg intravenously twice daily	10 mg quinine hydrochloride/kg IBW thrice daily starting 8 h after loading dose. After 48 h, 10 mg/kg IBW twice daily	7.3 (3.4; 9.1)	31 (45.6)
Quinine/Clindamycin	20 mg quinine hydrochloride (16.4 mg base)/kg IBW loading dose intravenously. Clindamycin: 600–900 mg IV thrice daily intravenously	10 mg quinine hydrochloride/kg IBW thrice daily starting 8 h after loading dose. After 48 h, 10 mg/kg IBW twice daily	11.3 (6.5; 18.6)	4 (5.9)

**Table 3 microorganisms-11-01579-t003:** Co-infections identified among 68 patients with imported falciparum malaria requiring intensive care.

Type of Co-Infection	Isolated Pathogen	Patients Affected, *n*
Community-acquired co-infections (*n* = 13)		
Chronic viral co-infection	HIV HCV	7 1
Mixed malaria	*P. malariae* + *P. falciparum*	2
Other travel-associated co-infection	*L. interrogans* Dengue virus	1 1
≥1 Co-infection	HBV + *P. vivax* + *P. falciparum*	1
Healthcare-associated co-infections (*n* = 8)		
Central-line-associated bloodstream infection (CLABSI)	*S. epidermidis*	3
Aspiration/pneumonia	*S. aureus* (MSSA) *S. aureus* (MSSA) and *P. aeruginosa*	1 1
Catheter-associated urinary tract infection (CAUTI)	*E. coli*	3

Abbreviations: HBV, hepatitis B; HCV, hepatitis C; HIV, human immunodeficiency virus; MSSA, methicillin-sensitive staphylococcus aureus.

**Table 4 microorganisms-11-01579-t004:** Association of the individual malaria-specific complications with the length of ICU stay in 68 patients requiring intensive care.

Criterion	Patients Affected, *n* (%)	Number of Total Malaria-Specific Complications on Admission, Median (IQR)	Median LOS-ICU (IQR)	Hazard Ratio (95%CI) for ICU Discharge by 38 h	*p*-Value	Hazard Ratio (95%CI) for ICU Discharge by 61 h	*p*-Value	Hazard Ratio (95%CI) for ICU Discharge by 91 h	*p*-Value
Jaundice	21 (30.9)	2 (1; 5)	64 (48; 122)	0.78 (0.44–1.39)	0.405	0.68 (0.33–1.38)	0.283	1.00 (0.37–2.67)	0.993
Hyper-parasitemia	18 (26.5)	2 (1; 4)	66 (40; 109)	0.84 (0.45–1.57)	0.586	0.77 (0.36–1.67)	0.515	0.98 (0.34–2.85)	0.968
Decompensated shock	12 (17.7)	4 (4; 6)	160(82; 332)	0.28 (0.13–0.60)	0.001	0.31 (0.13–0.73)	0.008	0.29 (0.09–0.98)	0.045
Renal impairment	12 (17.7)	5 (3; 6)	188 (109; 258)	0.288 (0.14–0.58)	<0.001	0.38 (0.18–0.80)	0.011	0.60 (0.22–1.64)	0.319
Respiratory distress (APO + ARDS)	11 (16.2)	4 (3; 6)	200 (146; 390)	0.17 (0.07–0.42)	<0.001	0.17 (0.06–0.44)	<0.001	0.21 (0.06–0.67)	0.009
APO	6 (8.8)	5 (3; 7)	146 (77; 192)	0.47 (0.19–1.21)	0.118	0.48 (0.17–1.38)	0.175	0.63 (0.18–2.23)	0.47
ARDS	5 (7.4)	4 (3; 4)	275 (238; 504)	0.24 (0.09–0.65)	0.005	0.28 (0.10–0.77)	0.013	0.41 (0.14–1.23)	0.111
Significant bleeding	10 (14.7)	4 (2; 6)	115 (67; 134)	0.52 (0.25–1.10)	0.084	0.63 (0.28–1.41)	0.262	0.91 (0.33–2.55)	0.858
Severe malarial anemia	10 (14.7)	4 (2; 6)	139 (64; 428)	0.30 (0.12–0.73)	0.008	0.32 (0.12–0.85)	0.023	0.17 (0.04–0.77)	0.021
Coma	7 (10.3)	6 (4; 7)	195 (109; 390)	0.36 (0.16–0.82)	0.016	0.45 (0.19–1.08)	0.073	0.70 (0.25–2.00)	0.51
Metabolic acidosis	7 (10.3)	6 (4; 7)	195 (124; 352)	0.32 (0.13–0.77)	0.011	0.40 (0.16–1.00)	0.050	0.58 (0.20–1.69)	0.317
Hypoglycemia	1 (1.5)	9 (9; 9)	195 (195; 195)	0.47 (0.19–1.21)	0.118	0.48 (0.17–1.38)	0.175	1.20 (0.15–9.56)	0.862
Convulsions	0 (0.0)	-	-	-	-	-	-	-	-

Abbreviations: APO, acute pulmonary oedema; ARDS, acute respiratory distress syndrome, according to [6].

**Table 5 microorganisms-11-01579-t005:** Results of the multivariate cox hazard proportional regression.

Covariate	Coefficient	Exponent of Coefficient (95%CI)	*p*-Value
Age	−0.011	0.989 (0.954; 1.026)	0.563
Origin from malaria-endemic country	−0.087	0.917 (0.307; 2.736)	0.876
Artemisinin therapy	0.862	2.368 (1.013; 5.538)	0.047
Severe malarial anemia	−0.169	0.845 (0.236; 3.023)	0.795
Shock	−0.737	0.479 (0.185; 1.239)	0.129
Respiratory distress	−1.427	0.240 (0.077; 0.751)	0.014

Abbreviations; 95%CI, 95% confidence interval.

**Table 6 microorganisms-11-01579-t006:** Univariate logistic regression for the factors associated with the development of respiratory distress among 68 patients with imported falciparum malaria requiring intensive care.

				Bayesian Logistic Regression				Firth’s Logistic Regression		
Covariate	Total (*n* = 68)	No Respiratory Complications (*n* = 57)	Respiratory Distress (*n* = 11)	Median of Posterior Distribution	95% Credibility Interval	Pd ^1^ (%)	% in ROPE ^2^	*p*-Value	Coefficient	Unadjusted OR (95% Confidence Interval)
Socio-demographic parameters										
Age, median (IQR), y	40 (31–52)	39 (31–53)	47 (37–51)	0.02	−0.03, 0.06	73.2	100	0.51	0.015	1.02 (0.97–1.06)
Male gender, *n* (%)	43 (63.2)	36 (63.2)	7 (63.6)	0.03	−1.27, 1.58	51.9	20.61	0.98	−0.018	0.98 (0.28–3.82)
Origin from malaria-endemic country, *n* (%)	35 (51.5)	29 (50.9)	6 (54.5)	0.16	−1.16, 1.44	59.60	22.37	0.84	0.133	1.14 (0.33–4.13)
History of ≥ 1 previous malaria episode, *n* (%)	9 (13.2)	8 (14.0)	1 (9.1)	−0.81	−3.78, 1.22	76.48	11.74	0.84	−0.184	0.83 (0.08–4.36)
Traveling as a tourist, *n* (%)	20 (29.4)	16 (28.1)	4 (36.4)	0.36	−1.14, 1.36	68.80	11.74	0.54	0.411	1.51 (0.38–5.45)
Duration from symptom onset to diagnosis ^3^, median (IQR), d	5 (3–7)	5 (3–7)	5 (3–5)	−0.18	−0.54, 0.14	85.60	50.82	0.34	−0.145	0.87 (0.61–1.15)
Co-morbidity										
No. of chronic co-morbidities, median (IQR)	0 (0–1)	0 (0–1)	1 (0–2)	0.46	−0.34, 1.22	88.45	18.55	0.19	0.483	1.62 (0.78–3.30)
CA-CCI, median (IQR)	0 (0–2)	0 (0–2)	2 (0–5)	0.44	0.11, 0.79	99.52	4.55	0.01	0.412	1.51 (1.10–2.13)
Diabetes (total *n*) *n* (%)	3 (4.4)	3 (5.3)	0 (0)	−6.61	−25.74, 0.67	94.83	1.47	0.79	−0.390	0.68 (0.01–7.72)
History of or active malignancy, *n* (%)	3 (4.4)	1 (1.8)	2 (18.2)	2.70	0.15, 6.08	97.97	0.13	0.04	2.294	9.91 (1.19–117.58)
Chronic pulmonary disease, *n* (%)	1 (1.5)	0 (0)	1 (9.1)	15.76	1.78, 49.32	99.25	0.0	0.07	2.80	16.43 (0.82–2465.97)
Chronic alcohol use, *n* (%)	4 (5.9)	4 (7.0)	0 (0)	−6.86	−23.25, 0.40	96.30	1.42	0.64	0.700	0.52 (0.00–5.40)
Obesity, *n* ^4^ (%)	8 (13.3)	7 (13.7)	1 (11.1)	−0.62	−3.91, 1.38	71.40	12.24	0.95	−0.065	0.94 (0.09–5.12)
BMI ^4^, kg/m^2^, median (IQR)	24.5 (23.0–26.8)	24.5 (23.2–27.1)	24.1 (22.7–24.5)	−0.08	−0.29, 0.11	76.80	85.13	0.52	−0.062	0.94 (0.76–1.13)
Any type of co-infection, *n* (%)	19 (27.9)	10 (17.5)	9 (81.8)	3.02	1.55, 4.77	100	0.0	<.001	2.844	17.2 (4.13–100.87)
HIV co-infection, *n* (%)	7 (10.3)	5 (8.8)	2 (18.2)	0.75	−1.35, 2.44	77.20	11.29	0.30	0.921	2.51 (0.40–12.38)
Malaria-specific complications										
Jaundice, *n* (%)	21 (30.9)	16 (28.1)	5 (45.5)	0.71	−0.64, 2.08	85.65	11.95	0.25	0.755	2.13 (0.58–7.68)
Hyper-parasitemia, *n* (%)	18 (26.5)	14 (24.6)	4 (36.4)	0.50	−0.97, 1.87	74.60	16.42	0.39	0.588	1.80 (0.45–6.58)
Decompensated shock, *n* (%)	12 (17.7)	5 (8.8)	7 (63.3)	2.95	1.45, 4.62	100	0.0	<0.001	2.767	15.90 (3.89–75.6)
Renal impairment, *n* (%)	12 (17.7)	5 (8.8)	7 (63.3)	2.89	1.33, 4.62	100	0.0	<0.001	2.767	15.90 (3.89–75.6)
Significant bleeding, *n* (%)	10 (14.7)	7 (12.3)	3 (27.3)	0.95	−0.76, 2.53	87.33	10.29	0.19	1.020	2.77 (0.58–11.66)
Severe malarial anemia, *n* (%)	10 (14.7)	5 (8.8)	5 (45.5)	2.20	0.66, 3.79	99.78	0	0.005	2.089	8.70 (1.93–35.71)
Coma, *n* (%)	7 (10.3)	2 (3.5)	5 (45.5)	3.27	1.46, 5.56	100	0.0	<0.001	2.933	18.79 (3.71–124.30)
Laboratory parameters										
Minimum base excess on admission ^5^, mmol/L, median (IQR)	−0.7 (−2.8–1.0)	−0.5 (−2.2–1.3)	−5.3 (−9.75– −3.6)	−0.35	−0.57, −0.16	100	2.5	<0.001	−0.305	0.74 (0.58–0.88)
Hypoglycemia, *n* (%)	1 (1.5)	0 (0)	1 (9.1)	16.27	1.60, 48.69	99.42	0.0	0.07	2.799	16.43 (0.82–2465.07)
Parasitemia, %, median (IQR)	8.0 (2.5–10.1)	8.0 (2.5–10.0)	8.0 (4.0–16.0)	0.06	−0.02, 0.14	92.90	100	0.12	0.058	1.06 (0.98–1.14)
Hemoglobin on admission, g/dL, median (IQR)	12.8 (11.1–13.9)	12.9 (11.4–13.7)	11.3 (8.7–14.2)	−0.15	−0.38, 0.09	89.22	62.37	0.22	−0.143	0.87 (0.69–1.09)
Blood urea on admission ^6^, mg/dL, median (IQR)	44 (28, 73)	41 (28–68)	66 (48–141)	0.02	0.0, 0.03	99.55	100	0.015	0.015	1.02 (1.00–1.03)
Hypoalbuminemia ^7^ (< 35 g/L), *n* (%)	25 (38.5)	17 (30.9)	8 (80.0)	2.15	0.69, 4.04	99.85	0.0	0.004	2.012	7.48 (1.83–42.82)
Disease severity on admission										
SAPS II score on admission ^8^, median (IQR)	32 (24–40)	26 (21–36)	42 (36–54)	0.09	0.04, 0.16	99.98	100	<0.001	0.08	1.09 (1.03–1.16)
Management										
Fluid intake day 1 ^9^, mL/kg/h, median (IQR)	2.2 (1.6–3.1)	2.1 (1.6–2.8)	2.7 (2.3–4.8)	0.80	0.26, 1.42	99.78	0.0	<0.001	0.720	2.04 (1.26–3.64)
Fluid intake day 2, mL/kg/h, median (IQR)	1.4 (.9–2.3)	1.1 (.8–1.8)	2.4 (2.1–3.6)	0.85	0.24, 1.60	99.78	0.0	0.006	0.750	2.11 (1.23–4.21)
Transfusion required, *n* (%)	11 (16.2)	5 (8.8)	6 (54.5)	2.57	1.13, 4.16	100	0.0	<0.001	2.423	11.3 (2.76–50.91)
RRT required, *n* (%)	7 (10.3)	2 (3.5)	5 (45.5)	3.21	1.49, 5.47	100	0.0	<0.001	2.933	18.79 (3.71–124.30)
Artemisinin-based regimen, *n* (%)	33 (48.5)	27 (47.4)	6 (54.5)	0.29	−0.99, 1.62	67.00	21.79	0.67	0.271	1.31 (0.37–4.75)

^1^ pd, probability of direction; ^2^ ROPE, region of practical equivalence; ^3^ data missing for nine cases; ^4^ data missing for seven cases; ^5^ data missing for seven cases; ^6^ data missing for two cases; ^7^ data missing for three cases; ^8^ data missing for 14 cases; ^9^ data missing for five cases.

**Table 7 microorganisms-11-01579-t007:** Results of the multivariate Bayesian logistic regression for the risk factors of respiratory distress among 68 patients with imported falciparum malaria requiring intensive care.

Parameter	Median, (95% CI)	OR (95% CI)	Pd (%)	% in ROPE (%)
Intercept	−5.48 (−8.87, −3.15)	-	100	0
Decompensated shock	2.44 (0.42, 4.73)	11.47 (1.52, 113.30)	99.10	0
Any type of co-infection	2.01 (0.15, 4.14)	7.46 (1.16, 62.80)	98.35	0
Fluid intake on day 1 (mL/kg/h)	0.77 (0.11, 1.63)	2.16 (1.12, 5.10)	98.72	0

Abbreviations: CI, credibility interval; OR, odds ratio; Pd, probability of direction; ROPE, region of practical equivalence.

## Data Availability

The data presented in this study are available on reasonable request only from the corresponding author. The data are not publicly available due to potential violation of privacy of enrolled patients.

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
