# Peer review of "Respiratory Distress Complicating Falciparum Malaria Imported to Berlin, Germany: Incidence, Burden, and Risk Factors"

_microorganisms, 2023, doi:10.3390/microorganisms11061579_

Round 1

Reviewer 1 Report

The present study examined a cohort of imported falciparum malaria cases treated in , well-equipped ICUs provided unlimited resources in terms of diagnostics, medication, organ support, and trained medical stuff familiar with the unique pathophysiology of the disease. In this setting respiratory distress proved to be the only malaria-specific complication independently associated with prolonged ICU-LOS. The study emphasizes the significance of controlling co-infections and avoiding fluid overload, even in shocked individuals, as these factors increase the risk of developing respiratory distress. However, after applying selection criteria the sample is too small and as the authors mention difficult to analice  statistically. Indeed, there are many different statistical strategies approach studies of this type.  

 my major concern is that is retrospective study thestudy designn and a long observation period, limiting its generalizability's. This inherent methodical problem was met by performing sensitivity analyses and by employing Bayesian logistic regression, an approach known to yield robust results even in small data sets. The odds ratios for the individual risk factors should be interpreted with caution.

 To many variables were considered however in my opinion the tables are full of data that are not necessary to include just those that statistically have some importance.

 conclusions must be revised.  focused only in those findings with really relevant.  

Author Response

Reply to Reviewer #1: First of all, I appreciate the efforts of the reviewers to provide their helpful reviews aiming to improve the quality of the manuscript. Many thanks for that!

Please find revisions according to remarks of Reviewer #1 highlighted in blue in the revised manuscript.

Remark #1: Thank you for this important remark. You are absolutely right: especially table 5 is a challenge for the reader. Yet, in my opinion it is critical to present all findings, significant or not, not least in order to provide a comprehensive list of all parameters that have been examined. For further reasons why reporting of non-significant results is important please refer to “The earnestness of being important: Reporting non-significant statistical results”; J Adv Nurs, 2020. For better clarity, sub-headings have now been added to Table 5 to give this confusing table more structure.

Remark #2: Thank you. The study aims to characterize incidence and burden of as well as risk factors for respiratory distress in imported malaria. The conclusions sections focuses only on these three key issues.

Reviewer 2 Report

The manuscript used existing data from imported P falciparum malaria cases to investigate the complications that are associated with prolonged ICU stay before the COVID-19 pandemic period.  Author has collected the extensive dataset, analyzed and presented it very comprehensively. The observational dataset is very well defined and very detailed analysis being conducted. I would only suggest to cite some of the relevant references (already cited in the main text, for example WHO definition of SM) in the tables as well.

Other critical suggestion is having this article edited for English grammar to enhance ease of understanding.

Author Response

Reply to Reviewer #2: First of all, I appreciate the efforts of the reviewers to provide their helpful reviews aiming to improve the quality of the manuscript. Many thanks for that!

Please find revisions according to remarks of Reviewer #2 highlighted in red in the revised manuscript.

Remark #1: Thanks for this important remark. It alerted me that I forgot to cite a publication of critical importance for the manuscript: the “Berlin definition” of ARDS. As recommended, key citations have been added to the tables where appropriate.

Remark #2: Thank you. Since I am not a native speaker I have to apologize for erroneous grammar. Once more, I have gone through the manuscript to make corrections. Fortunately, the journal provides excellent English revision during the publication process.